# ResGAT: A Residual Graph Attention Network for Cancer Subtype Classification in Whole Slide Images

**Zhenhan Lin** [1]                                                    ZHENHAN.LIN@VANDERBILT.EDU

**Hao Tong** [3]                                                         TONGH@ALUMNI.WFU.EDU

**Yunfei Hu** [1]                                                        YUNFEI.HU@VANDERBILT.EDU

**Xianyong Gui** [4]                                                 XIANYONG.GUI@WFUSM.EDU

**Jeanne Shen** [5]                                                   JEANNES@STANFORD.EDU

**Byrne Lee** [6]                                                       BYRNELEE@STANFORD.EDU

**Lu Zhang** [7]                                                        ERICLUZHANG@HKBU.EDU.HK

**Daniel Moyer** [1]                                                  DANIEL.MOYER@VANDERBILT.EDU

**Mu Zhou** [8]                                                         MUZHOU1@GMAIL.COM

**Xin Maizie Zhou** [1,2,*]                                        MAIZIE.ZHOU@VANDERBILT.EDU

**Konstantinos Votanopoulos** [3,*]                      KVOTANOP@WAKEHEALTH.EDU

[1] *Department of Computer Science, Vanderbilt University, Nashville, TN, United States*

[2] *Department of Biomedical Engineering, Vanderbilt University, Nashville, TN, United States*

[3] *Department of General Surgery, Wake Forest University, Winston-Salem, NC, United States*

[4] *Department of Pathology, Wake Forest University, Winston-Salem, NC, United States*

[5] *Department of Pathology, Stanford University School of Medicine, Palo Alto, CA, United States*

[6] *Department of Surgery, Stanford University, Palo Alto, CA, United States*

[7] *Department of Computer Science, Hong Kong Baptist University, Hong Kong*

[8] *Department of Computer Science, Rutgers University, New Brunswick, NJ, United States*

**Editors:** Under Review for MIDL 2026

## Abstract

Multiple instance learning (MIL) provides a weakly supervised framework for whole slide image (WSI) classification, enabling slide-level prediction from gigapixel images with only slide-level labels. However, WSI subtype classification in realistic settings is still challenging. In this work, we propose ResGAT, a residual graph attention framework that operates on hybrid $k$-NN patch graphs and models WSI representations with stacked residual graph attention blocks. ResGAT is evaluated on the subtype classification task across a rare, class-imbalanced appendiceal cancer cohort, BRACS and two TCGA datasets. It outperforms SOTA MIL baselines on the appendiceal cancer and BRACS cohorts, and remains competitive on the TCGA datasets. On the appendiceal cancer cohort, we further assess cross-site generalization via few-shot adaptation under source shift, showing that ResGAT adapts effectively to new domains with limited labels. An ablation study is provided to validate the effectiveness of key architectural components of our method.

**Keywords:** whole slide image classification, multiple instance learning, residual graph attention framework, cross-site generalization

## 1. Introduction

As histopathology digitization becomes routine, incorporating computational models into diagnostic workflows is increasingly feasible (Hanna et al., 2019; Kumar et al., 2020;

Zhang et al., 2025). These computational models provide slide-level classification results together with interpretable justifications, promoting consistent decisions and transparent verification (Tizhoosh and Pantanowitz, 2018; Yilmaz et al., 2024). This is particularly valuable for rare diseases, where expert diagnosticians are scarce. However, a fundamental challenge lies in the gigapixel scale of whole-slide images (WSIs), which prevents them from being processed as a single image. In practice, the standard approach involves tiling tissue regions into thousands of patches, formulating the task as a Multiple Instance Learning (MIL) problem.

The evolution of MIL for WSI classification has shifted from simple feature pooling to sophisticated context modeling. Initial frameworks adopted static aggregation strategies, such as max-pooling (Campanella et al., 2019) and mean-pooling. While computationally efficient, these methods often lose critical contextual information by focusing only on the extreme feature or diluting signals through averaging. The introduction of Attention-based MIL (ABMIL) (Ilse et al., 2018) marked a pivotal advancement by using trainable weights to rank instances. Subsequent research has sought to address overfitting and attention concentration through advanced strategies: pseudo-bag augmentation and feature distillation methods like DTFD-MIL (Zhang et al., 2022); and attention-challenging frameworks such as ACMIL (Zhang et al., 2024) and MHIM (Tang et al., 2023) that mitigate attention concentration by suppressing high-confidence instances to encourage the discovery of comprehensive diagnostic patterns. Despite these improvements, the attention mechanisms often treat instances as independent and identically distributed (i.i.d.). To explicitly capture inter-instance correlations, recent sequence-based works like TransMIL (Shao et al., 2021) and the Mamba-based architecture (Yang et al., 2024) leverage self-attention and selective scan mechanisms to explicitly model long-range dependencies, marking a paradigm shift towards correlated feature learning.

Running parallel to sequence-based advancements, Graph Neural Networks (GNNs) have emerged as a distinct paradigm focused on explicitly encoding the structural topology of the tissue (Brussee et al., 2025). By representing patches as nodes and their interactions as edges, these methods avoid flattening the spatial structure into a sequence. Early implementations employed $k$-nearest neighbor ($k$NN) algorithms to construct spatial graphs, demonstrating that explicitly modeling local neighborhoods enhances diagnostic accuracy (Chen et al., 2021; Zheng et al., 2022). Subsequent research has explored more intricate graph constructions, including hierarchical formulations for multi-resolution reasoning (Hou et al., 2022) and heterogeneous graphs that distinguish between different tissue components (Chan et al., 2023). However, the "over-smoothing" phenomenon (Chen et al., 2020) is challenging for graph-based MIL approaches. Stacking multiple message passing layers induces node representations to become homogenized, losing the discriminative power essential for classification. This degradation poses an obstacle in realistic clinical settings, which are characterized by extreme heterogeneity in tissue scale. In such diverse scenarios, the fact that applying standard readout functions to homogenized features yields inconsistent diagnostic profiles across varying graph sizes, harming the reliability required for clinical deployment.

Motivated by these challenges, we propose ResGAT, a weakly supervised MIL framework for whole slide image subtype classification. The whole slide image is represented as a hybrid $k$-NN patch graph with nodes initialized by extracted patch features and connected

via spatial and feature proximity. ResGAT processes the patch graphs with stacked residual graph attention blocks, where each block features a dual-branch design combining multi-head graph attention with a parallel linear projection. This design preserves patch-specific information while adaptively aggregating contextual information, yielding representations that support effective slide-level prediction. In comparative evaluations against representative MIL baselines, our model achieves superior classification performance on both a rare, class-imbalanced appendiceal cancer cohort and the multi-class BRACS dataset, while it remains competitive on two public TCGA datasets. On the appendiceal cancer cohort, we also introduce a benchmarking protocol to assess cross-site generalization and few-shot adaptation, demonstrating that ResGAT maintains strong performance when labeled data are limited in new domains. An ablation study is provided to examine the effectiveness of the core components of ResGAT. Furthermore, the framework supports qualitative interpretation through heatmaps that highlight prediction-relevant regions.

## 2. Method

### 2.1. Problem Formulation

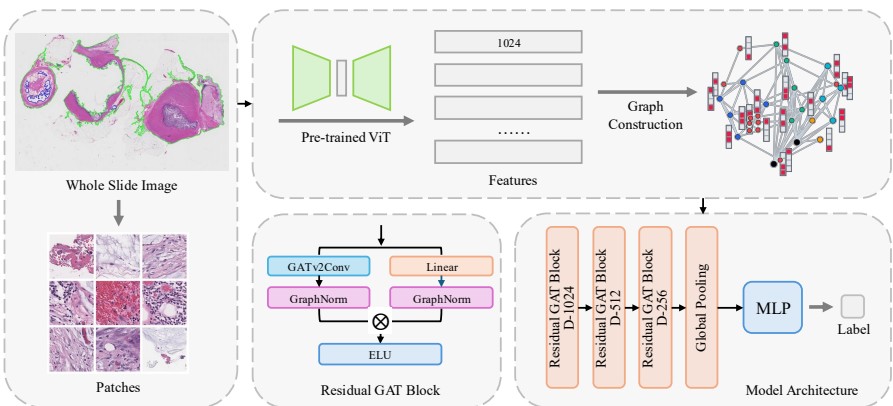

Figure 1: Overview of the ResGAT pipeline for WSI classification. The framework consists of three main components: (1) tissue segmentation and patch extraction, (2) patch-level feature encoding and graph construction, and (3) slide-level representation learning and prediction.

The whole slide image is treated as a bag of patch embeddings in the multiple instance learning (MIL) setting. Given a slide $s$, the foreground tissue is segmented and tiled into patches at a fixed magnification. Each patch is then encoded into a feature vector $\mathbf{x}_i \in \mathbb{R}^D$ using a large-scale pre-trained pathology encoder. This yields a set $\mathcal{B}_s = \{\mathbf{x}_1, \ldots, \mathbf{x}_N\}$ with a slide-level label $y_s$ indicating the cancer subtype. Our goal is to learn a permutation-invariant function $f_\theta : \mathcal{B}_s \mapsto y_s$ for subtype classification.

Following previous graph-based MIL methods, we represent each slide as a patch graph $\mathcal{G}_s = (\mathcal{V}_s, \mathcal{E}_s)$. Each node $v_i \in \mathcal{V}_s$ corresponds to a patch embedding $\mathbf{x}_i$, and the edges in $\mathcal{E}_s$

are constructed based on both spatial proximity and feature similarity between the patches. ResGAT takes the graph as input, updates node features with stacked residual graph attention blocks, and aggregates them into a slide-level representation for classification. Fig. 1 shows the overall architecture of ResGAT.

## 2.2. Graph Construction

To establish the graph topology $\mathcal{E}_s$, we introduce a hybrid $k$-NN edge construction procedure. Each node $v_i$ is associated with a spatial coordinate $\mathbf{p}_i \in \mathbb{R}^2$ derived from the patch location on the WSI. Initially, we identify the $d_{spa}$ nearest spatial neighbors of $v_i$ measured by Euclidean distance between coordinates, denoted as the set $\mathcal{N}_{spa}(v_i)$, and its $d_{feat}$ nearest feature neighbors measured by cosine distance between node features, denoted as the set $\mathcal{N}_{feat}(v_i)$. We define the candidate pool as the intersection

$$\mathcal{C}(v_i) = \mathcal{N}_{spa}(v_i) \cap \mathcal{N}_{feat}(v_i),$$

which is subsequently ranked by node feature similarity. The top $k$ candidates are selected as the final connected neighbors of $v_i$. In cases of a sparse or empty intersection ($|\mathcal{C}(v_i)| < k$), the adjacency list is padded with up to three auxiliary nearest feature neighbors to maintain robust connectivity. The resulting patch graph $\mathcal{G}_s$ is treated as undirected. The hyperparameters $d_{spa}, d_{feat}, k$ jointly determine the graph density and the node degree variance. We adopt a general configuration with $k = 6, d_{feat} = 50, d_{spa} \in \{15, 24\}$ in our main evaluations. A comprehensive sensitivity analysis of these parameters is provided in Section 3.4.1.

## 2.3. ResGAT Architecture and Training Objective

**Node Updates.** Given the graph $\mathcal{G}_s$, we initialize the node features as $\mathbf{h}_i^{(0)} = \mathbf{x}_i$ for $i = 1, ..., N$. Let $\mathbf{h}_i^{(\ell)}$ denote the representation of node $v_i$ at layer $\ell$. ResGAT applies a stack of $L = 3$ residual blocks to obtain the updated node representations $\mathbf{h}_i^{(L)}$. Each residual block updates node features through a linear projection in parallel with a multi-head graph attention convolution (GATv2Conv (Brody et al., 2021)). Let $\mathcal{N}(i) = \{j \mid (i, j) \in \mathcal{E}_s\}$ denote the neighbors of node $v_i$. For each layer, the following combined update is applied to all nodes:

$$
\begin{aligned}
e_{ij}^{(k)} &= \mathbf{a}^{(k)\top} \text{LeakyReLU}\big(\mathbf{W}_s^{(k)}\mathbf{h}_i^{(\ell)} + \mathbf{W}_t^{(k)}\mathbf{h}_j^{(\ell)}\big), \quad j \in \mathcal{N}(i), \\
\alpha_{ij}^{(k)} &= \frac{\exp(e_{ij}^{(k)})}{\sum_{u \in \mathcal{N}(i)} \exp(e_{iu}^{(k)})}, \\
\mathbf{m}_i^{(\ell)} &= \Big\|_{k=1}^{K} \sum_{j \in \mathcal{N}(i)} \alpha_{ij}^{(k)} \mathbf{W}^{(k)} \mathbf{h}_j^{(\ell)}, \\
\mathbf{h}_i^{(\ell+1)} &= \phi\Big(\text{GN}\big(\mathbf{m}_i^{(\ell)}\big) + \text{GN}\big(\mathbf{W}_{\text{res}}^{(\ell)}\mathbf{h}_i^{(\ell)}\big)\Big),
\end{aligned}
\tag{1}
$$

where $d_h = D_{\ell+1}/K$ is the output dimension of each attention head, $\mathbf{W}_s^{(k)}, \mathbf{W}_t^{(k)}, \mathbf{W}^{(k)} \in \mathbb{R}^{d_h \times D_\ell}$ are learnable projections for head $k$, $\mathbf{a}^{(k)} \in \mathbb{R}^{d_h}$ is the corresponding attention vector,

and $\mathbf{W}_{\text{res}}^{(\ell)} \in \mathbb{R}^{D_{\ell+1} \times D_\ell}$ is the learnable linear projection on the residual path. The operator $\|$ denotes concatenation over $K$ heads. $\text{GN}(\cdot)$ denotes GraphNorm and is applied separately to the two branches, and $\phi$ is the ELU non-linearity. This formulation accommodates progressively decreasing dimensions (e.g., $1024 \to 512 \to 256$).

**Graph Normalization.** Each residual block employs GraphNorm (Cai et al., 2021) to stabilize training against the severe variations in graph size and topological structure across different slides. Given the intermediate node representations at the layer $\ell$, GraphNorm defines the operation as

$$\mathbf{u}_i^{(\ell)} = \boldsymbol{\gamma} \odot \frac{\mathbf{f}_i^{(\ell)} - \boldsymbol{\alpha} \odot \boldsymbol{\mu}^{(\ell)}}{\sqrt{\left(\boldsymbol{\sigma}^{(\ell)}\right)^2 + \epsilon}} + \boldsymbol{\beta}, \tag{2}$$

where $\boldsymbol{\mu}^{(\ell)}$ and $\left(\boldsymbol{\sigma}^{(\ell)}\right)^2$ are the mean and variance of $\{\mathbf{f}_i^{(\ell)}\}_{i=1}^N$ over nodes in the graph, and $\boldsymbol{\gamma}, \boldsymbol{\beta}, \boldsymbol{\alpha}$ are learnable parameters shared across nodes. The operator $\odot$ denotes element-wise multiplication. Intuitively, $\boldsymbol{\gamma}$ and $\boldsymbol{\beta}$ provide a channel-wise affine re-parametrization of the normalized features, while $\boldsymbol{\alpha}$ modulates the strength of graph-level centering on each feature dimension.

**Pooling and Loss.** Following residual blocks, we apply the global mean pooling over the updated node representations $\{\mathbf{h}_i^{(L)}\}_{i=1}^N$ to obtain the slide-level representation $\mathbf{z}_s \in \mathbb{R}^{D_L}$. This vector is fed into an MLP classifier to produce the logit vector $\hat{\mathbf{y}}_s \in \mathbb{R}^C$, where $C$ is the number of cancer subtypes. The predicted probabilities are obtained via a Softmax function. Given the ground-truth label encoded as a one-hot vector $\mathbf{y}_s \in \{0,1\}^C$, we train the model using the standard cross-entropy loss:

$$\mathcal{L} = -\frac{1}{|\mathcal{S}|} \sum_{s \in \mathcal{S}} \sum_{c=1}^C y_{s,c} \log\left(\frac{\exp(\hat{y}_{s,c})}{\sum_{c'=1}^C \exp(\hat{y}_{s,c'})}\right). \tag{3}$$

An ablation study of the two-branch residual block design is provided in Section 3.4.2.

### 2.4. Graph Class Activation Mapping

We adapt Grad-CAM++ (Chattopadhay et al., 2018) to our graph-based pipeline to generate heatmaps that highlight prediction-relevant regions. Given a target class $c$, let $h_{i,d}^{(L)}$ denote the $d$-th feature channel of the final node representation $\mathbf{h}_i^{(L)} \in \mathbb{R}^{D_L}$. We compute channel-wise importance weights $w_d^c$ from the gradients of the class logit $\hat{y}_c$:

$$w_d^c = \alpha_d^c \cdot \frac{1}{N} \sum_{i=1}^N \text{ReLU}\left(\frac{\partial \hat{y}_c}{\partial h_{i,d}^{(L)}}\right), \qquad \alpha_d^c = \frac{\sum_i \left(\frac{\partial \hat{y}_c}{\partial h_{i,d}^{(L)}}\right)^2}{2\sum_i \left(\frac{\partial \hat{y}_c}{\partial h_{i,d}^{(L)}}\right)^2 + N\sum_i \left(\frac{\partial \hat{y}_c}{\partial h_{i,d}^{(L)}}\right)^3 + \epsilon}. \tag{4}$$

The saliency score $M_i^c$ for each node is computed via a weighted combination:

$$M_i^c = \text{ReLU}\left(\sum_{d=1}^{D_L} w_d^c h_{i,d}^{(L)}\right). \tag{5}$$

These scores are min-max normalized and mapped back to the corresponding patch locations on the WSI.

## 3. Experiments

### 3.1. Dataset and Experimental Setup

#### 3.1.1. Dataset

**Appendiceal cancer cohort.** This cohort consists of 141 diagnostic WSIs of 92 patients with low-grade appendiceal mucinous neoplasm (LAMN) and mucinous adenocarcinoma (MAC). It is significantly imbalanced (LAMN:MAC = 32:15). Sourcing from both Wake Forest and Stanford introduces additional domain shift challenges. Clinically, MAC is regarded as the more aggressive subtype with worse prognosis than LAMN, so it is treated as the positive class when computing AUC and the reported F1-score corresponds to the positive label.

**TCGA datasets.** Two public datasets, NSCLC and ESCA, were curated from The Cancer Genome Atlas (TCGA) program (Tomczak et al., 2015). Both datasets involve distinguishing adenocarcinoma from squamous cell carcinoma: LUAD vs LUSC for NSCLC, and EAC vs ESCC for the esophagus ESCA. For evaluation, the clinically more aggressive squamous cell carcinoma was treated as the positive class when computing the AUC.

**BRACS dataset.** The BRACS dataset (Brancati et al., 2022) is a public dataset with 526 WSIs of various breast lesions. It contains seven diagnostic categories: normal (N), benign lesions (PB), usual ductal hyperplasia (UDH), atypical ductal hyperplasia (ADH), flat epithelial atypia (FEA), ductal carcinoma in situ (DCIS), and invasive carcinoma (IC). Due to severe class imbalance, both the AUC and F1-score were computed using macro-averaging across all categories.

Slide counts per diagnostic category are summarized in Table 6 in the Appendix. All tissue segmentation and patch extraction were performed at 20× magnification.

#### 3.1.2. Evaluation protocols

All experiments were conducted on an NVIDIA RTX A6000 GPU with 48GB memory. For feature extraction, we adopted the CLAM (Lu et al., 2021) preprocessing pipeline with HSV-based tissue segmentation and contour-based spatial sampling to identify tissue regions. Patch-level 1024-dimensional feature vectors were extracted using UNI (Chen et al., 2024)(ViT-L/16 via DINOv2) with standard ImageNet normalization (Deng et al., 2009).

For each cohort, we performed slide-level 5-fold cross-validation with patient-wise splits. In each split, three folds were used for training, one for validation and one for testing. We report the mean and standard deviation of metrics over the five test folds. Specifically, balanced accuracy is reported for the highly imbalanced Appendiceal cancer and BRACS cohorts, while overall accuracy is used for the TCGA cohorts.

For the domain adaptation analysis on the appendiceal cancer cohort, WF slides formed the source domain and SF slides the target domain. The WF data were partitioned into training, validation and test subsets in a 70/15/15 ratio for pre-training each model. For the target domain, we defined a fixed SF test set of 12 slides (10 LAMN and 2 MAC); this SF test set was used for all zero-shot and few-shot evaluations. Zero-shot performance was obtained

Table 1: Subtype classification performance (mean$_{std}$, %) across four datasets: appendiceal cancer, TCGA-NSCLC, TCGA-ESCA, and BRACS. Results are reported as balanced accuracy (BAcc), accuracy, AUC, F1 and F1-macro.

| Method | Appendiceal Cancer | | | BRACS | | |
|---|---|---|---|---|---|---|
| | BAcc | AUC | F1 | BAcc | AUC | F1-macro |
| CLAM-SB | $90.09_{6.47}$ | $94.96_{8.79}$ | $86.25_{8.15}$ | $31.25_{4.19}$ | $77.29_{3.21}$ | $27.11_{4.45}$ |
| CLAM-MB | $88.62_{10.68}$ | $\mathbf{96.82_{4.13}}$ | $85.36_{14.95}$ | $30.48_{3.30}$ | $74.72_{3.12}$ | $27.56_{3.89}$ |
| DSMIL | $78.92_{13.86}$ | $90.58_{9.89}$ | $68.44_{24.77}$ | $32.15_{6.74}$ | $68.44_{3.66}$ | $29.31_{6.84}$ |
| TransMIL | $84.07_{10.71}$ | $92.47_{7.64}$ | $76.87_{14.33}$ | $30.18_{4.07}$ | $76.09_{2.57}$ | $26.80_{3.89}$ |
| WiKG | $84.31_{7.39}$ | $94.37_{6.51}$ | $79.16_{11.19}$ | $28.08_{1.86}$ | $70.75_{2.93}$ | $22.58_{2.42}$ |
| PatchGCN | $87.41_{9.48}$ | $95.35_{7.42}$ | $83.84_{12.29}$ | $28.93_{3.85}$ | $71.57_{4.68}$ | $21.87_{5.11}$ |
| DTFD-MIL | $86.22_{9.56}$ | $93.27_{11.35}$ | $80.08_{13.02}$ | $26.82_{3.63}$ | $73.37_{2.10}$ | $21.36_{3.48}$ |
| MHIM-DSMIL | $86.42_{12.74}$ | $97.03_{2.72}$ | $81.15_{19.45}$ | $33.29_{7.18}$ | $77.47_{3.89}$ | $\mathbf{31.64_{5.95}}$ |
| MHIM-TransMIL | $87.49_{9.45}$ | $91.59_{11.69}$ | $84.94_{13.47}$ | $27.16_{2.17}$ | $68.89_{3.55}$ | $25.32_{1.53}$ |
| **ResGAT (ours)** | $\mathbf{92.56_{6.36}}$ | $96.41_{1.94}$ | $\mathbf{90.98_{7.98}}$ | $\mathbf{33.76_{6.07}}$ | $\mathbf{77.61_{0.95}}$ | $28.74_{6.08}$ |

| Method | TCGA-NSCLC | | TCGA-ESCA | |
|---|---|---|---|---|
| | Accuracy | AUC | Accuracy | AUC |
| CLAM-SB | $\mathbf{93.72_{1.72}}$ | $\mathbf{97.55_{1.44}}$ | $\mathbf{98.04_{1.60}}$ | $99.83_{0.34}$ |
| CLAM-MB | $92.70_{1.53}$ | $97.39_{1.57}$ | $96.11_{3.16}$ | $\mathbf{100.00_{0.00}}$ |
| DSMIL | $92.29_{1.40}$ | $97.08_{1.53}$ | $95.42_{4.45}$ | $97.72_{2.65}$ |
| TransMIL | $92.29_{2.13}$ | $97.15_{0.82}$ | $93.51_{4.08}$ | $99.39_{0.52}$ |
| WiKG | $92.09_{1.94}$ | $96.35_{1.45}$ | $93.48_{3.59}$ | $99.63_{0.74}$ |
| PatchGCN | $93.00_{2.08}$ | $97.13_{1.53}$ | $92.84_{2.37}$ | $99.17_{1.45}$ |
| DTFD-MIL | $93.61_{1.75}$ | $97.41_{1.38}$ | $96.11_{3.16}$ | $99.39_{0.65}$ |
| MHIM-DSMIL | $92.70_{1.23}$ | $97.48_{1.23}$ | $94.82_{4.37}$ | $98.88_{1.80}$ |
| MHIM-TransMIL | $92.40_{1.31}$ | $97.30_{1.51}$ | $94.82_{4.37}$ | $99.73_{0.22}$ |
| **ResGAT (ours)** | $93.51_{0.75}$ | $97.15_{1.47}$ | $98.02_{1.62}$ | $99.91_{0.17}$ |

by applying the WF-pretrained model directly to the SF test set. For few-shot adaptation, we fine-tuned the pretrained model on small labeled SF subsets with 3, 6 and 9 training slides and separate validation sets of 3, 3 and 5 slides, respectively. After adaptation, we report overall accuracy on the SF test set and quantify adaptation efficacy using backward transfer (BWT) and forward transfer (FWT). We use overall accuracy in this setting rather than balanced accuracy to provide a clearer view of adaptation trends. BWT is defined as the change in WF test accuracy before and after fine-tuning, where large negative BWT values indicate catastrophic forgetting. FWT is computed as the improvement of SF test accuracy over the zero-shot baseline, where positive values indicate successful adaptation.

See Appendix A for more implementation details.

### 3.2. Comparison with state-of-the-art methods

We compared our method with nine strong MIL baselines that cover diverse design paradigms: attention-based pooling MIL (CLAM-SB and CLAM-MB (Lu et al., 2021)), transformer-based MIL (TransMIL (Shao et al., 2021)), dual-stream MIL (DSMIL (Li et al., 2021)), distillation-based MIL (DTFD-MIL (Zhang et al., 2022)), graph-based MIL

Table 2: Domain adaptation performance comparison. Source refers to pre-trained test accuracy from WF dataset. Zero-shot refers to SF test performance on two classes data separately without adaptation. FWT measures forward transfer (target improvement), BWT measures backward transfer (source performance retention). Class 0 and Class 1 represent LAMN and MAC respectively.

| Method | Source(WF) Accuracy | Zero-shot (SF) class 0 | class 1 | 3-shot (SF) Acc | FWT↑ | BWT↑ | 6-shot (SF) Acc | FWT↑ | BWT↑ | 9-shot (SF) Acc | FWT↑ | BWT↑ |
|---|---|---|---|---|---|---|---|---|---|---|---|---|
| WiKG | 89.47 | 100 | 0 | 83.33 | 0 | 10.53 | 83.33 | 0 | 5.26 | 83.33 | 0 | 5.26 |
| TransMIL | 84.21 | 100 | 0 | 83.33 | 0 | 0 | 83.33 | 0 | 0 | 83.33 | 0 | 0 |
| DSMIL | 73.68 | 70 | 50 | 75.0 | 8.33 | 0 | 75.0 | 8.33 | 0 | 75.0 | 8.33 | 0 |
| MHIM-DSMIL | 84.21 | 90 | 0 | 75.0 | 0 | 5.26 | 75.0 | 0 | 5.26 | 83.33 | 8.33 | 0 |
| MHIM-TransMIL | 89.47 | 100 | 0 | 83.33 | 0 | 0 | 91.67 | 8.33 | 5.26 | 100 | 16.67 | 5.26 |
| CLAM-MB | 89.47 | 100 | 0 | 83.33 | 0 | 0 | 83.33 | 0 | 0 | 83.33 | 0 | 5.26 |
| CLAM-SB | **94.74** | 90 | 0 | 75.0 | 0 | 0 | 75.0 | 0 | 0 | 75.0 | 0 | 0 |
| DTFD-MIL | 89.47 | 90 | 100 | 91.67 | 0 | 0 | 100 | 8.33 | 0 | 100 | 8.33 | 5.26 |
| PatchGCN | 91.67 | **100** | **100** | 100 | 0 | 0 | 100 | 0 | 0 | 100 | 0 | 2.64 |
| **ResGAT** | **92.86** | 100 | 50 | **91.67** | **8.33** | **0** | **100** | **8.33** | **0** | **100** | **8.33** | **0** |

(WiKG (Li et al., 2024) and PatchGCN (Chen et al., 2021)), and hard-instance-mining MIL (MHIM-DSMIL and MHIM-TransMIL (Tang et al., 2023, 2026)). Table 1 reports mean and standard deviation over five folds for all metrics on the four datasets.

On the appendiceal cancer cohort, ResGAT achieves the highest balanced accuracy at 92.56±6.36%, outperforming the best baseline CLAM-SB by roughly 2.5% and yielding the lowest standard deviation across folds. It also attains the highest F1-score and a high AUC, indicating reliable detection of the clinically more aggressive MAC subtype. On TCGA-NSCLC and TCGA-ESCA, CLAM-SB attains the highest mean accuracy, while ResGAT remains competitive: its accuracy is only 0.21% and 0.02% below CLAM-SB on TCGA-NSCLC and TCGA-ESCA, respectively. Notably, ResGAT's low standard deviations on TCGA cohorts shows stable performance across folds. On BRACS, a challenging seven-class fine-grained classification task, ResGAT achieves the highest balanced accuracy and AUC among all methods, while MHIM-DSMIL obtains the highest F1-macro. The overall low balanced accuracy across all methods reflects the inherent difficulty of fine-grained breast lesion subtyping. Overall, these results indicate that ResGAT performs well on the class-imbalanced and label-noisy appendiceal cancer cohort and the BRACS dataset, while remaining comparable to competitive MIL baselines on other datasets.

The results also highlight the complementary strengths of other MIL approaches. On the two TCGA cohorts, DTFD-MIL obtains the second highest accuracies and AUCs, with CLAM-MB generally close behind. The MHIM variants (MHIM-DSMIL and MHIM-TransMIL) consistently improve over their backbones, and show the effectiveness of the hard-instance mining strategy.

### 3.3. Domain Adaptation Analysis

In this experiment, we evaluated cross-site robustness on the appendiceal cancer cohort, where WF and SF correspond to different acquisition sites (see Section 3.1.1 for details). Such cross-site settings often introduce substantial distribution shift due to differences in

scanners, staining protocols and local practice, and models trained on a single site can experience a marked performance drop when deployed elsewhere (Liu et al., 2025; PoceviVCiute et al., 2024). We therefore used this scenario to assess generalization ability of methods, which is a critical consideration for realistic clinical deployment. We first evaluated zero-shot performance, where a model trained on the source site is directly applied to the target site. Subsequently, we evaluate few-shot adaptation, where only a small number of labeled SF slides are available for finetuning the source-trained model (see Section 3.1.2 for details).

### 3.3.1. CROSS-DOMAIN GENERALIZATION

Table 2 compares our method with the same nine MIL baselines. While most MIL baselines achieve reasonably high accuracy on the WF source test set, their zero-shot performance on the SF target set is highly variable and often subtype-imbalanced. Several baselines, including WiKG, TransMIL and the CLAM variants, fail to correctly predict MAC samples during cross-site transfer, indicating a strong bias toward the majority subtype when crossing sites. In comparison, PatchGCN and DTFD-MIL achieve strong zero-shot performance on the SF test set, with per-class accuracies exceeding 90%, suggesting robust initial cross-site generalization. ResGAT achieves the second-highest source-domain accuracy on the WF test set and provides competitive zero-shot accuracy on the SF test set, establishing a solid foundation for further adaptation.

### 3.3.2. FEW-SHOT ADAPTATION

In this setting, we analyzed how pre-trained models adapt to target data when finetuned on a small number of labeled SF slides. ResGAT demonstrates superior adaptation efficiency, reaching 100% overall accuracy on the SF test set at the 3-shot setting and maintaining this performance across the 6-shot and 9-shot settings. Its already high source test performance remains robust across all settings (BWT = 0), showing that adaptation does not induce forgetting on the source domain. This result suggests that ResGAT can be effectively adapted to a new site with only a small number of labeled slides, which is especially valuable in rare-disease scenarios where annotation is costly and limited.

PatchGCN maintains its perfect accuracy throughout all few-shot settings, suggesting that its graph-based representation captures site-invariant tissue structures. DTFD-MIL and MHIM-TransMIL show steady improvements on SF test accuracy as more target slides become available, alongside positive forward transfer (FWT) and BWT, indicating stable learning and knowledge retention under additional target supervision. By contrast, CLAM-SB and CLAM-MB, despite their strong performance on general classification tasks, show little change in SF test accuracy across all few-shot settings, suggesting that their architectures are less responsive to limited target supervision during cross-site adaptation.

## 3.4. Ablation Study

### 3.4.1. EFFECTIVENESS OF PROPOSED EDGE CONSTRUCTION

To assess the contribution of the proposed graph construction, we compared several topology variants: Feature kNN (edges based on feature similarity), Spatial kNN (edges

Table 3: Ablation on graph construction. Values are reported as mean$_{std}$ over 5-fold cross-validation (%).

| Graph Variant | Appendiceal Cancer | | TCGA-NSCLC | | TCGA-ESCA | | BRACS | |
| | BAcc | AUC | Accuracy | AUC | Accuracy | AUC | BAcc | AUC |
| --- | --- | --- | --- | --- | --- | --- | --- | --- |
| Feature kNN | $90.97_{4.82}$ | $96.45_{3.86}$ | $92.70_{1.71}$ | $97.21_{1.17}$ | $98.02_{1.62}$ | $99.91_{0.17}$ | $32.30_{2.55}$ | $77.13_{1.27}$ |
| Spatial kNN | $91.79_{6.22}$ | $96.06_{4.22}$ | $93.10_{1.24}$ | $97.35_{1.37}$ | $97.35_{2.49}$ | $99.45_{0.68}$ | $33.41_{3.49}$ | $77.49_{1.70}$ |
| Hybrid ($d_{spa}$=24) | $\mathbf{92.56_{6.36}}$ | $\mathbf{96.41_{1.94}}$ | $92.60_{1.56}$ | $\mathbf{97.65_{0.90}}$ | $\mathbf{98.02_{1.62}}$ | $\mathbf{99.91_{0.17}}$ | $\mathbf{33.76_{6.07}}$ | $\mathbf{77.61_{0.95}}$ |
| Hybrid ($d_{spa}$=15) | $90.78_{6.05}$ | $94.23_{3.69}$ | $\mathbf{93.51_{0.75}}$ | $97.15_{1.47}$ | $98.02_{1.62}$ | $99.54_{0.93}$ | $31.54_{2.58}$ | $77.00_{2.09}$ |
| Node-permuted | $92.46_{6.14}$ | $96.08_{3.53}$ | $92.80_{0.99}$ | $97.70_{1.00}$ | $98.02_{1.62}$ | $99.83_{0.21}$ | $33.53_{3.98}$ | $76.95_{3.98}$ |

based on spatial proximity), Hybrid (our method with two settings of $d_{spa}$) and Node-permuted (hybrid adjacency with features randomly reassigned to nodes). For all graph variants, we use $k = 6$; for the hybrid case, we vary the $d_{spa}$ hyperparameter while keeping all other settings fixed (see Section 2.2 for details). As shown in Table 3, the hybrid graph consistently provides the strongest overall performance across datasets, indicating that combining spatial proximity and feature similarity yields a more effective graph topology than using either criterion alone. Notably, the node-permuted variant remains competitive. This suggests that the adjacency structure itself provides structural regularization that stabilizes representation learning and mitigates overfitting.

We further investigated the robustness of the hybrid topology through a sensitivity analysis of its hyperparameters. Specifically, we performed a grid search over the number of spatial neighbors ($d_{spa} \in \{15, 24, 36, 48, 60\}$), feature neighbors ($d_{feat} \in \{35, 50, 65, 90, 105\}$), and the maximum neighborhood size ($k \in \{6, 8\}$). The resulting heatmap in Appendix Fig. 2 visualizes the evaluation metrics across all four datasets. ResGAT exhibits stability over a broad spectrum of parameter combinations. Although the best parameter choices vary by dataset, the general configuration consistently provides strong performance across all datasets. Overall, these results demonstrate that the proposed edge construction is effective and robust across diverse datasets.

### 3.4.2. Effectiveness of Residual Block Design

We conducted a set of experiments to evaluate key architectural design choices, including dual-branch architecture, normalization strategy, layer depth, and graph convolution type. First, we compared the performance of the full ResGAT model against two variants: one ablating the linear branch and another removing all inter-node edges, which degenerates the model into a node-wise MLP. As shown in Table 4, ablating the linear branch leads to a consistent drop in performance across datasets, indicating that direct patch-level feature propagation meaningfully complements graph aggregation. Disconnecting the inter-node edges also resulted in a decline in accuracy and balanced accuracy. Together, these findings confirm that the dual-branch architecture is essential. Preserving patch-specific features and aggregating topological context are both important for forming effective slide-level representations. Table 5 shows that GraphNorm provides the most favorable performance within ResGAT compared to LayerNorm and InstanceNorm. Specifically, it outperforms both alternatives on the appendiceal cancer and BRACS datasets, where its graph-level

Table 4: Ablation on the linear branch and inter-node edges. Values are reported as mean$_{std}$ over 5-fold cross-validation (%).

| Setting | Appendiceal Cancer | | TCGA-NSCLC | | TCGA-ESCA | | BRACS | |
|---|---|---|---|---|---|---|---|---|
| | BAcc | AUC | Acc | AUC | Acc | AUC | BAcc | AUC |
| w/o Inter-node Edges | $88.95_{6.71}$ | $\mathbf{96.52_{3.08}}$ | $93.11_{0.59}$ | $\mathbf{97.94_{0.74}}$ | $97.38_{2.43}$ | $99.82_{2.20}$ | $31.29_{2.89}$ | $75.34_{1.63}$ |
| w/o Linear Branch | $87.08_{6.77}$ | $93.79_{3.56}$ | $92.70_{1.60}$ | $97.08_{1.66}$ | $93.51_{3.49}$ | $99.29_{0.60}$ | $29.41_{4.11}$ | $74.66_{3.89}$ |
| **ResGAT** | $\mathbf{92.56_{6.36}}$ | $96.41_{1.94}$ | $\mathbf{93.51_{0.75}}$ | $97.15_{1.47}$ | $\mathbf{98.02_{1.62}}$ | $\mathbf{99.91_{0.17}}$ | $\mathbf{33.76_{6.07}}$ | $\mathbf{77.61_{0.95}}$ |

Table 5: Ablation on normalization layers in ResGAT. Values are mean$_{std}$ over 5-fold cross-validation (%).

| Normalization | Appendiceal Cancer | | TCGA-NSCLC | | TCGA-ESCA | | BRACS | |
|---|---|---|---|---|---|---|---|---|
| | BAcc | AUC | Accuracy | AUC | Accuracy | AUC | BAcc | AUC |
| InstanceNorm | $89.23_{7.70}$ | $95.84_{2.19}$ | $\mathbf{93.51_{0.66}}$ | $\mathbf{97.18_{1.41}}$ | $98.02_{1.62}$ | $99.91_{0.17}$ | $33.45_{2.33}$ | $76.03_{1.91}$ |
| LayerNorm | $81.32_{11.15}$ | $91.31_{7.19}$ | $91.48_{1.98}$ | $96.63_{1.77}$ | $93.46_{4.08}$ | $99.30_{0.57}$ | $25.22_{3.96}$ | $69.76_{4.77}$ |
| GraphNorm | $\mathbf{92.56_{6.36}}$ | $\mathbf{96.41_{1.94}}$ | $93.51_{0.75}$ | $97.15_{1.47}$ | $\mathbf{98.02_{1.62}}$ | $\mathbf{99.91_{0.17}}$ | $\mathbf{33.76_{6.07}}$ | $\mathbf{77.61_{0.95}}$ |

normalization statistics and learnable shift parameter offer a more expressive normalization strategy than the per-node or per-feature counterparts.

Additionally, Appendix Table 7 summarizes the impact of different layer depths and graph convolution types. For the layer depth study, the 2-layer variant removes the intermediate residual block with appropriate dimension alignment, while the 4-layer variant adds an additional block that preserves the output dimension of the third layer. The results show that a 3-layer configuration provides the best trade-off between performance and computational cost: 2-layer models generally underperform due to limited receptive fields, whereas increasing the depth to 4 layers incurs higher computational cost without yielding consistent improvements. Regarding the graph convolution type, we compared GAT against GCN, GIN, and GraphSAGE. While GIN performs comparably in specific instances, we adopt GAT as the default because it achieves the highest performance across the majority of metrics and remains the most stable choice across datasets.

We further evaluated computational efficiency by comparing the throughput of ResGAT against two other graph-based methods, WiKG and Patch-GCN, under the same training protocol. As detailed in Appendix Table 8, ResGAT achieves a throughput comparable to Patch-GCN and WiKG, demonstrating that the multi-layer residual block design enhances performance without incurring significant computational overhead.

### 3.5. Qualitative Results

We applied graph-adapted Grad-CAM++ (Section 2.4) to visualize WSI heatmaps. Appendix Fig. 3 illustrates three representative MAC cases, where the primary heatmaps in the first row were computed as a confidence-weighted average of the top cross-validation models. The regions with high saliency scores are outlined in yellow. While heatmaps from individual folds exhibit spatial variation, the top-performing models show consensus in the regions

they highlight, indicating that the network captures stable diagnostic patterns. Clinical review by our pathologist confirmed that the high-scoring patches predominantly correspond to tumor and stromal tissue, where tumor morphology and its spatial relationship with the stroma inform subtyping. This indicates that ResGAT's predictions are informed by histologically meaningful features.

## 4. Conclusion

In this work, we propose ResGAT, a graph-based MIL framework for WSI subtype classification. The architecture features a dual-branch residual graph attention design that preserves patch-specific features while adaptively aggregating graph-based context, helping mitigate the feature homogenization commonly associated with standard message passing. Our ablation study further shows that this dual-branch structure is effective, with direct patch-level feature propagation meaningfully complementing graph aggregation. Additionally, this study reveals that the proposed hybrid kNN graph topology, together with GraphNorm and a 3-layer GAT configuration, contributes to the overall performance of ResGAT. Our main results demonstrate that ResGAT outperforms SOTA MIL baselines on the class-imbalanced, label-noisy appendiceal cancer cohort and the challenging multi-class BRACS dataset, while remaining competitive on TCGA-NSCLC and TCGA-ESCA datasets. To assess model robustness under realistic deployment conditions, we introduce a cross-site evaluation protocol on the appendiceal cancer cohort that measures zero-shot generalization and few-shot adaptation across acquisition sites. In this setting, ResGAT reaches full target-site accuracy with only a few labeled slides and without forgetting the source domain. Notably, several MIL methods that perform well in general classification task fail to adapt under the cross-site setting. These observations suggest that, while general benchmarking provides a valuable baseline, extending evaluations to realistic diagnostic settings gives a more complete picture of a model's clinical efficacy.

## Acknowledgments

This work was supported by the NIGMS Maximizing Investigators' Research Award (MIRA) R35 GM146960.

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

## Appendix A. Implementation Details

For ResGAT model, we trained for 30 epochs using Adam optimizer with learning rate $3 \times 10^{-4}$ and weight decay $1 \times 10^{-4}$. To account for randomness, each experiment was repeated with two random seeds 3 and 3407; the best-performing run is reported. Following standard MIL practice, we applied batch size of 1. For baseline methods, we used their recommended hyperparameters from official implementations to ensure fair comparison.

## Appendix B. Dataset Table

Table 6: Dataset statistics for Appendiceal Cancer, TCGA-NSCLC, TCGA-ESCA and BRACS. WSIs passing quality control are included. For the TCGA cohorts, slides with missing or ambiguous histologic labels were excluded.

| Dataset | Label | Diagnosis | Number of WSIs Site 1 | Site 2 |
|---------|-------|-----------|-----------------------|--------|
| Appendiceal Cancer | 0 | LAMN | 74 | 22 |
| | 1 | MAC | 40 | 5 |
| TCGA-NSCLC | 0 | LUAD | 496 | |
| | 1 | LUSC | 490 | |
| TCGA-ESCA | 0 | EAC | 63 | |
| | 1 | ESCC | 90 | |
| BRACS | 0 | N | 34 | |
| | 1 | PB | 142 | |
| | 2 | UDH | 70 | |
| | 3 | ADH | 46 | |
| | 4 | FEA | 41 | |
| | 5 | DCIS | 61 | |
| | 6 | IC | 132 | |

# Appendix C. Supplementary Results

Table 7: Ablation study on layer depth (left) and graph convolution type (right) across different datasets. All results use the same evaluation protocol as the main results.

| Metric | Number of Layers | | | Graph Convolution Type | | | |
|---|---|---|---|---|---|---|---|
| | 2 | 3 | 4 | GCN | GIN | SAGE | GAT |
| **Appendiceal Cancer** | | | | | | | |
| **BAcc** | $90.44_{5.08}$ | $\mathbf{92.56_{6.36}}$ | $88.09_{7.97}$ | $88.05_{8.07}$ | $91.02_{5.03}$ | $89.53_{4.47}$ | $\mathbf{92.56_{6.36}}$ |
| **AUC** | $94.56_{3.73}$ | $\mathbf{96.41_{1.94}}$ | $95.83_{2.12}$ | $93.39_{3.39}$ | $\mathbf{97.45_{1.63}}$ | $96.16_{2.31}$ | $96.41_{1.94}$ |
| **F1** | $87.51_{5.81}$ | $\mathbf{90.98_{7.98}}$ | $86.31_{10.53}$ | $83.76_{11.67}$ | $87.79_{5.84}$ | $85.57_{7.00}$ | $\mathbf{90.98_{7.98}}$ |
| **TCGA-ESCA** | | | | | | | |
| **Accuracy** | $\mathbf{98.04_{2.59}}$ | $98.02_{1.62}$ | $98.02_{1.62}$ | $96.73_{2.04}$ | $96.09_{2.40}$ | $98.02_{1.62}$ | $\mathbf{98.02_{1.62}}$ |
| **AUC** | $99.64_{0.44}$ | $\mathbf{99.91_{0.17}}$ | $99.82_{0.22}$ | $99.72_{0.56}$ | $99.74_{0.51}$ | $\mathbf{100.00_{0.0}}$ | $99.91_{0.17}$ |
| **TCGA-NSCLC** | | | | | | | |
| **Accuracy** | $92.39_{1.35}$ | $93.51_{0.75}$ | $\mathbf{94.01_{1.63}}$ | $91.58_{1.60}$ | $93.51_{2.33}$ | $91.99_{2.27}$ | $\mathbf{93.51_{0.75}}$ |
| **AUC** | $97.31_{1.05}$ | $97.15_{1.47}$ | $\mathbf{97.69_{1.15}}$ | $97.51_{0.99}$ | $\mathbf{97.97_{0.85}}$ | $97.10_{1.37}$ | $97.15_{1.47}$ |
| **BRACS** | | | | | | | |
| **BAcc** | $31.77_{6.18}$ | $\mathbf{33.76_{6.07}}$ | $31.95_{1.66}$ | $33.26_{3.03}$ | $\mathbf{34.01_{2.67}}$ | $33.76_{5.33}$ | $33.76_{6.07}$ |
| **AUC** | $76.16_{2.36}$ | $77.61_{0.95}$ | $\mathbf{77.67_{0.73}}$ | $76.30_{0.88}$ | $76.26_{0.92}$ | $75.95_{3.12}$ | $\mathbf{77.61_{0.95}}$ |
| **F1-macro** | $28.06_{6.00}$ | $28.74_{6.08}$ | $\mathbf{29.14_{2.45}}$ | $26.77_{2.57}$ | $\mathbf{32.10_{2.25}}$ | $29.93_{4.91}$ | $28.74_{6.08}$ |

Table 8: Computational efficiency comparison of three graph-based WSI methods on the same data splitting. Throughput is measured as the average number of graph batches processed per second (approximately WSIs per second) over a 30-epoch run.

| Method | Average Throughput |
|---|---|
| Patch-GCN | 7.03 |
| WiKG | 5.40 |
| ResGAT | 6.91 |

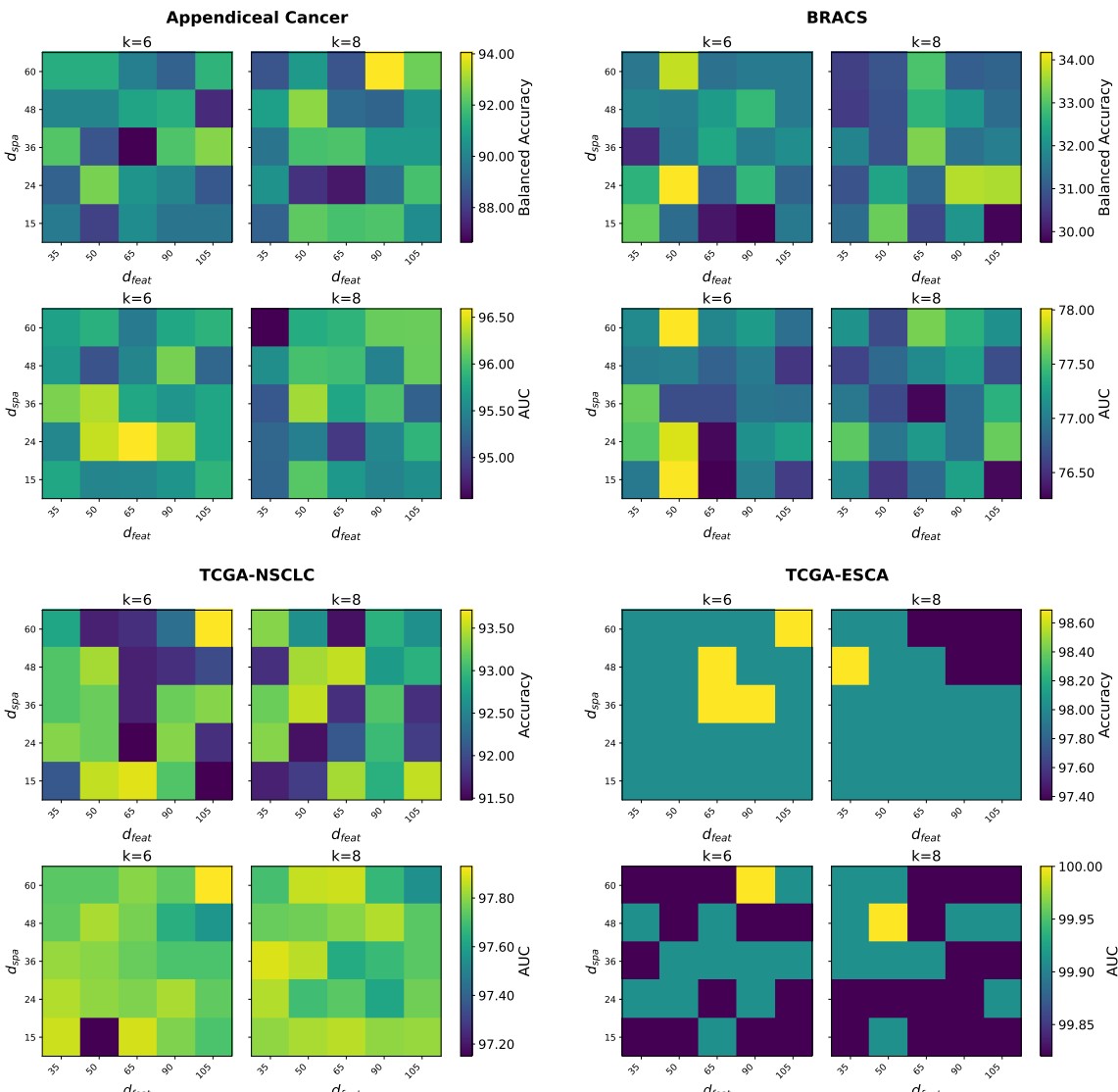

Figure 2: Hyperparameter sensitivity heatmaps across four datasets (Appendiceal Cancer, BRACS, TCGA-NSCLC, TCGA-ESCA). For each dataset, we visualize the performance over the hyperparameters $(d_{spa},\ d_{feat})$ grid at two graph sparsity settings $(k = 6, 8)$. The top row reports the primary metric (Balanced Accuracy for Appendiceal Cancer and BRACS datasets; Accuracy for others), and the bottom row reports AUC score. Within each dataset, the two heatmaps in the same row share the colorbar to enable direct comparison between $k$ values; brighter colors indicate better performance.

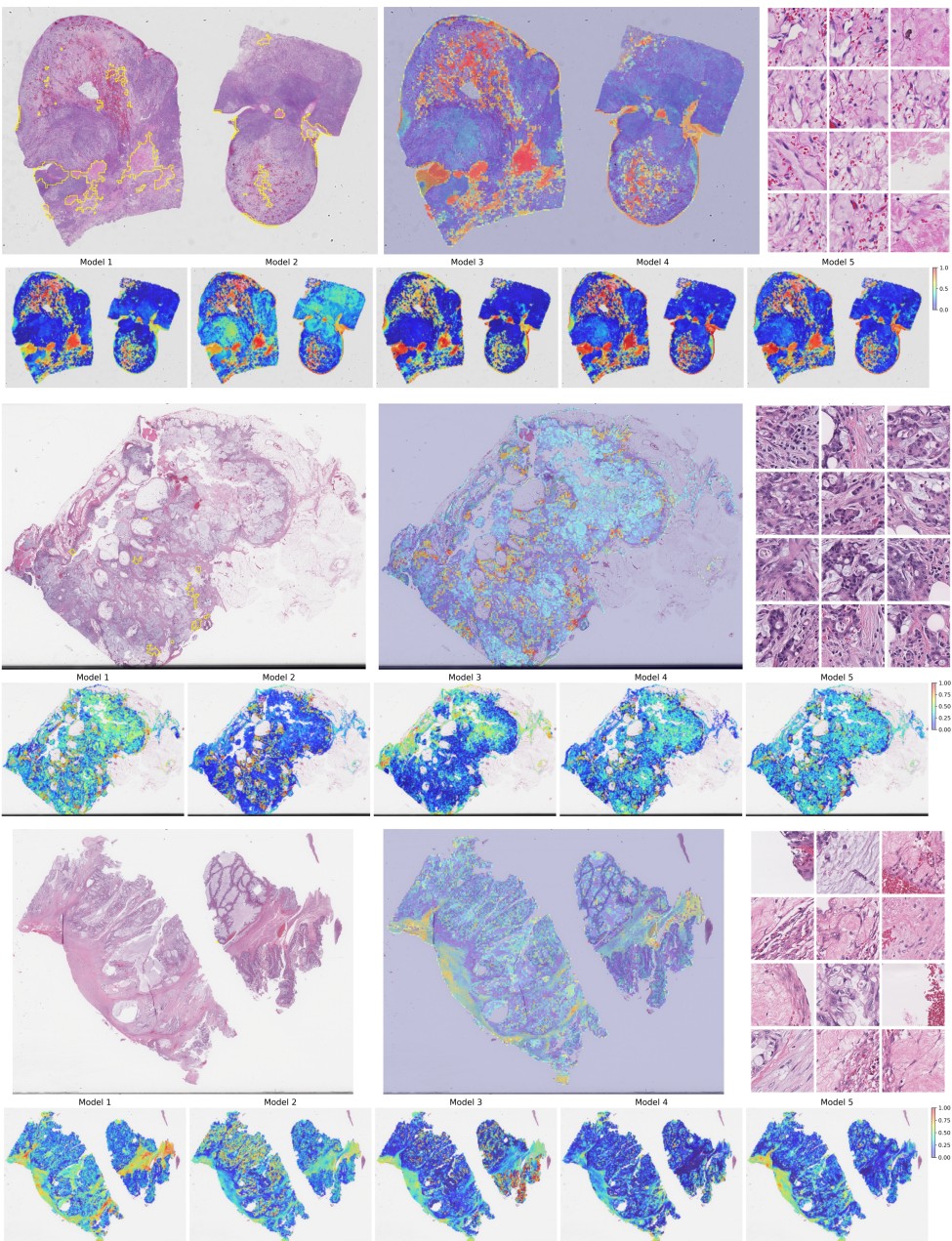

Figure 3: Heatmap visualizations for representative MAC cases (WF53, S22, S36). The first row shows the aggregated heatmap and the corresponding high-contribution regions outlined in yellow, computed as a confidence-weighted average of Models 1, 2, and 5. The second row displays heatmaps from the five cross-validation models. Selected high-contribution patches are shown for localized inspection.

