# OpenReview forum: "ResGAT: A Residual Graph Attention Network for Cancer Subtype Classification in Whole Slide Images"
_MIDL.io/2026/Conference — MIDL 2026 Poster_

### Official Review · Reviewer_vF2K · 2025-12-24

**Confidence:** 5
**Preliminary Rating:** 3
**Final Rating:** 4

**Summary:**

This paper proposes ResGAT, a residual graph attention network specifically designed for subtype classification of WSIs. The authors conducted experiments on a private dataset and two public datasets, showing that ResGAT outperforms existing baselines on the private dataset and achieves comparable performance on the public datasets. Ablation studies were also performed to investigate the effectiveness of the relevant modules.

**Strengths:**

1. The paper has a clear structure, and the methodology section is described in detail.
2. A wide range of baselines are used for comparison, and the experiments are comprehensive.
3. The proposed ResGAT performs excellently in few-shot domain transfer tasks.

**Weaknesses:**

1. ResGAT performs best on private datasets, but its performance on public datasets is not optimal.  Given the larger size of the public datasets, there is a concern that the current results may be subject to randomness, raising doubts about their statistical significance.
2. Why is a combination of residual connections and GAT necessary? What would the results be without residual connections? What about using other graph convolution methods such as GCN, GIN, and Sage? The paper does not include these comparative experiments, which weakens the motivation for the proposed method. Meanwhile, the paper mentions that challenges still exist in WSI subtyping tasks, but the three datasets presented are all binary classification tasks with already good performance. I believe that using a better-performing foundation model as a feature extractor (e.g., Virchow-v2, UNI-v2, etc.) would further improve its capabilities. It is suggested that the authors conduct experiments on more challenging subtyping tasks, such as EBRAINS[1] and BRACS[2] (which can be accessed in [3]), to explore its capabilities.
3. What is the computational efficiency of ResGAT? Given that it uses three GAT layers, can it be compared with other graph-based models, such as WiKG and Patch-GCN[4], to explore its computational efficiency?
4. The heatmap in the paper does not show any useful information. Please supplement the pathological characteristics of the high-scoring regions.

[1] Roetzer-Pejrimovsky T, Moser A C, Atli B, et al. The digital brain tumour atlas, an open histopathology resource[J]. Scientific Data, 2022, 9(1): 55.

[2] Brancati N, Anniciello A M, Pati P, et al. Bracs: A dataset for breast carcinoma subtyping in h&e histology images[J]. Database, 2022, 2022: baac093.

[3] Zhang A, Jaume G, Vaidya A, et al. Accelerating data processing and benchmarking of ai models for pathology[J]. arXiv preprint arXiv:2502.06750, 2025.

[4] Chen R J, Lu M Y, Shaban M, et al. Whole slide images are 2d point clouds: Context-aware survival prediction using patch-based graph convolutional networks[C]//International Conference on Medical Image Computing and Computer-Assisted Intervention. Cham: Springer International Publishing, 2021: 339-349.

**Detailed Comments:**

1.  Add ablation studies regarding the number of layers and compare the efficiency with common graph-based models.
2.  Add experiments on multi-class classification tasks to further explore the capabilities of ResGAT.
3.  Ablation studies on residual layers and graph convolution need to be provided.

**Justification Of Final Rating:**

Thank you to the author for their efforts in the rebuttal stage. All my concerns and problems have been completely addressed after the revised manuscript, and I have increased my score to weak accept.

**Justification Of The Preliminary Rating:**

Graph-based MIL methods are not uncommon in WSI tasks and have already shown good results in binary classification tasks for cancer typing. If the authors can address the concerns mentioned above, this would be a decent paper, and the score would be adjusted accordingly.

**Questions To Address In The Rebuttal:**

1. Provide results on multi-class classification tasks.
2. Concerns regarding the ablation studies of residuals and graph convolutions need to be addressed.
3. Comparison of computational efficiency, and supplementary details of the heatmap modeling experiment.

---

> ### Author Response · Authors · 2026-01-25
>
> We appreciate the reviewer for the careful and insightful evaluation and for the constructive suggestions aimed at strengthening the empirical validation of our work. In response to these comments, we have conducted additional experiments and analyses in the revised paper (see Supporting Material) to address concerns. Below, we provide point-by-point responses to the reviewer’s questions.
>
> > **Q1:** Evaluation on challenging multi-class benchmarks (BRACS) and validation of statistical significance.
>
> We sincerely appreciate the reviewer’s suggestion to evaluate ResGAT on more challenging, multi-class tasks to verify its robustness. We acknowledge the concern that results on small datasets may be subject to randomness. In our study, the private appendiceal cohort and TCGA-ESCA (~155 WSIs) both involve limited numbers of cases, which makes the classification task itself challenging and reflects realistic clinical data scarcity. In this setting, we note ResGAT performs best on the private cohort, while on public benchmarks it achieves results comparable to the strongest baselines, with only marginal differences (<0.4%).
>
> Following your recommendation, we conducted benchmarks on the BRACS dataset which presents a substantially more challenging multi-class problem with 7 classes, 526 WSIs, and severe class imbalance. For this benchmark, we utilized UNI-v1 features and set hyperparameters to $d_{neighbors}=24$, $f_{neighbors}=50$, and $k=6$, while keeping other implementation details consistent. Regarding the foundation model, while we acknowledge the potential of newer models like Virchow-v2 and UNI-v2, we retained UNI-v1 as it offers top-tier performance among 1024-dimensional models. (see Table 2 in revised paper) ResGAT achieves leading performance with the highest Balanced Accuracy (33.76%) and AUC (77.61%). These results show that ResGAT maintains competitive performance under a more challenging multi-class setting, highlighting its potential beyond binary classification tasks. We apologize for not including the EBRAINS dataset due to the limited rebuttal time window.
>
> > **Q2**: Ablation studies on model architecture: graph convolutions, residual connections, and layer Depth.
>
> We appreciate the reviewer’s rigorous inquiry into our architectural choices. To validate our architectural design, we conducted comprehensive ablation studies across all three datasets (see Table 8 in revised paper).
>
> + Optimal Layer Depth ($L=3$): Our experiments demonstrate that a 3-layer configuration offers the optimal trade-off between performance and computational efficiency. As observed in the tables, 2-layer models generally underperform due to limited receptive fields. Conversely, increasing the depth to 4 layers incurs higher computational costs without significant performance gains and, in the case of the private dataset, leads to performance degradation likely due to overfitting.
>
> + Superiority of GATv2Conv: We compared our GAT-based backbone against GCN, GIN, and GraphSAGE. The results show that the proposed ResGAT (GAT with residuals) consistently achieves the highest Accuracy and Balanced Accuracy across datasets. While other methods like GIN achieve competitive AUC scores in specific cases, ResGAT delivers the most stable and superior overall performance, confirming that the attention mechanism is effectively weighting diverse patch features in WSI subtyping.
> + Necessity of Residual Connections (see Table 6 in the revised paper): Removing residual connections inferior classification performance compared to the full ResGAT model. This validates their role in stabilizing the training of deeper graph networks and ensuring efficient feature propagation without incurring major computational costs.
>
> > **Q3**: Computational efficiency comparison
>
> We evaluated the computational efficiency of ResGAT by comparing it with WiKG and Patch-GCN, under the same training protocol. Efficiency is measured in terms of average throughput, defined as the number of graph batches (approximately WSIs) processed per second. The results (see Table 9 in the revised paper) show that graph-based WSI models operate at comparable throughput levels. Our proposed multi-layer residual GAT design does not introduce a significant computational overhead in practice.
>
> > **Q4**: Heatmap modeling experiment
>
> We would like to clarify that the heatmaps are primarily intended to visualize prediction-relevant regions identified by the model, rather than for definitive clinical diagnosis or explicit tumor localization. In the revised manuscript, we provided additional heatmap visualizations(AppendixD), including cross-validation heatmaps, aggregated heatmaps, and representative high-contribution patches. Qualitatively, we observe that the high-scoring regions frequently include tumor and stromal tissue components, suggesting that the model tends to emphasize histologically informative areas related to subtype characterization.

---

### Official Review · Reviewer_nTj7 · 2026-01-07

**Confidence:** 4
**Preliminary Rating:** 2
**Final Rating:** 2

**Summary:**

The paper addresses whole-slide image (WSI) subtype classification, a challenging problem due to the gigapixel scale of WSIs. The authors propose ResGAT, a residual graph attention framework that represents a WSI as a graph of patches, with edges defined by spatial and feature similarities. The model processes hybrid k-NN patch graphs using graph attention blocks followed by subtype classification, and is evaluated on three datasets.

**Strengths:**

1. Literature review: The paper provides a well-structured review of existing WSI classification methods, clearly outlining their clinical motivations (e.g., diagnosis verification and rare subtype detection) and technical limitations, including signal dilution in static aggregation, attention collapse in attention-based MIL, and over-smoothing in graph-based MIL approaches.

2. Experimental protocol: The evaluation is thorough, combining fixed train/validation/test splits with k-fold cross-validation across multiple datasets.

3. Methodology: The proposed graph connectivity strategy jointly leverages spatial proximity and feature similarity to model patch-level relationships in WSIs.

**Weaknesses:**

1. The proposed aggregation module builds upon existing graph attention mechanisms with an additional residual branch. These design choices may be perceived as incremental rather than fundamentally novel.

2. Organization: Key implementation details and qualitative results are deferred to the appendix, whereas they would benefit from inclusion in the main paper body, in line with typical MIDL submission expectations.

3. Ablation study: The analysis of k-NN graph hyperparameters is restricted (e.g., spatial parameter only, with few values), raising concerns about sensitivity to dataset-specific choices and the generalizability of the approach across different pathology settings.

**Detailed Comments:**

1. Table presentation: In Table 1, explicitly indicating best results in bold (if space permits) would improve readability. Similarly, adding down/up arrows for BWT and FWT in Table 2 would help non-expert readers quickly interpret these metrics and better highlight the advantages of the proposed method.

2. The paper would benefit from a more comprehensive ablation study of the graph construction hyperparameters (both d and f). Visualizing performance trends as a function of these parameters would improve clarity and help disentangle architectural design choices from hyperparameter optimization. Including a separate leave-one-out ablation study would ease readability and highlights the importance of each model component: i) without hybrid edge construction, ii) without GATV2 branch, and iii) without linear branch.

**Justification Of Final Rating:**

I thank the authors for their detailed response, the clarifications regarding the manuscript, and the additional experiments provided. In this work, the authors present ResGAT, a residual graph attention-based model for Whole Slide Image classification.

The experimental evaluation is thorough, including multiple datasets (both internal and external), strong baselines, and a rigorous validation protocol, ultimately demonstrating competitive performance. These results are further complemented by ablation studies on the main architectural components.

From a methodological perspective, ResGAT primarily builds upon existing frameworks (graph attention networks combined with a parallel linear projection), making the proposed graph construction strategy the central contribution of the paper. The hybrid kNN approach, which combines spatial and feature-based neighborhood definitions, is well motivated; however, its empirical analysis remains limited. The impact of this design choice is mainly assessed in a single table, where the hybrid strategy yields small but consistent AUROC improvements across datasets.

Overall, the paper would benefit from a more in-depth investigation of the proposed graph construction. In particular, additional empirical evidence supporting the claimed benefits beyond overall performance, such as improved training stability or robustness, would strengthen the contribution. Furthermore, an analysis of computational complexity and efficiency (e.g., effects on training or inference time due to restricted neighborhoods) would help clarify the practical advantages of the proposed approach.

**Justification Of The Preliminary Rating:**

The paper proposes ResGAT, an attention-based framework with a parallel linear projection branch for WSI subtype classification, using graph connectivity defined by spatial and feature similarity. The method is evaluated on three datasets and includes cross-dataset experiments. While the experimental validation and protocol are robust, the methodological contributions largely build on existing mechanisms, which may limit perceived novelty. Additionally, the analysis of graph construction hyper-parameters is limited, and key implementation details and qualitative results are relegated to the appendix, potentially affecting reproducibility and depth of evaluation.

**Questions To Address In The Rebuttal:**

1. Parallel branch: The introduction claims that the parallel linear projection path preserves individual patch information. How does this differ from standard self-loops or residual connections already present in graph attention layers (e.g., as implemented in PyTorch Geometric)?

2. Ablation study: Why is the ablation analysis focused on the spatial neighbor parameter (d) only, while feature-based neighbors (f) or the overall k in hybrid k-NN graphs are not explored (Table 4)?

3. Evaluation metrics: What is the rationale for reporting accuracy in Table 2, rather than balanced accuracy or F1-score, which may be more appropriate for potentially imbalanced subtype classification tasks?

---

> ### Author Response · Authors · 2026-01-25
>
> We thank the reviewer for the detailed and constructive feedback. In the revised paper, we have conducted additional experiments and analyses to address these concerns (see Supporting Material). Our responses to the specific questions are provided below.
>
> > **Q1:** Leave-one-out ablation and parallel branch clarification.
>
> We appreciate the suggestion regarding a clearer leave-one-out ablation analysis. We note that investigations for (i) *without hybrid edge construction* and (ii) *without the GATv2 branch* are already evaluated in the main paper (see Table 5 in the revised paper), where disconnected, feature-only, spatial-only, and hybrid graph variants are explicitly compared. We additionally report (iii) *without the parallel linear branch* (see Table 6 in the revised paper), which shows a consistent performance drop relative to the full ResGAT.
>
> Regarding the parallel linear branch, we would like to clarify how it differs from standard self-loops or residual connections implemented in PyTorch Geometric.
>
> + Unlike self-loops that implicitly retain node features within message passing, the linear branch explicitly preserves identical per-patch representations by bypassing neighborhood aggregation, as supported by Table 6.
> + The graph branch and linear branch in ResGAT are normalized independently and combined after normalization, whereas standard residual implementations share normalization. This design improves robustness under strong inter-slide heterogeneity by stabilizing gradient flow when graph aggregation becomes unstable.
>
> > **Q2:** Hyperparameter sensitivity problem
>
> We sincerely thank the reviewer for this constructive suggestion and extended our ablation study. To rigorously evaluate the sensitivity of  the hyperparameters, we conduct a grid search over spatial neighbors ($d_{neighbors} [15, 24, 36, 48, 60]$), feature neighbors ($f_{neighbors} [35, 50, 65, 90, 105]$), and graph size ($k [6, 8]$). We visualized the results using heatmaps (see Figure 2 in the revised paper). The results can be summarized as follows:
>
> + **Performance Stability**: ResGAT's performance remains stable across different parameter combinations without exhibiting strict trends. It consistently maintains higher accuracy than the best baseline method on the Appendiceal Cancer dataset across the majority of the grid.
> + **Rationale for Configuration**: Although our grid search identified specific combinations that yield higher performance than baselines, we adopted the "generalist" configuration($k=6$, $d_{neighbors}=14/25$,$f_{neighbors=50}$) used in the main paper. This choice is further supported by its strong performance on the newly benchmarked BRACS dataset (see Table 2 in revised paper).
>
> > **Q3:** Evaluation metrics
>
> Regarding the evaluation metrics in Table 2 (Table 3 in revised paper), accuracy is used to characterize adaptation behavior. In this domain adaptation setting, our primary focus is on whether target performance improves after few-shot fine-tuning (FWT) and whether source performance is preserved (BWT). For zero-shot evaluation, we reported per-class accuracies to expose subtype imbalance and majority-class bias. Following our adaptation protocol, a fixed SF test set (10 LAMN and 2 MAC slides) is used under which balanced accuracy or F1-score can be unstable. Balanced accuracy and F1-score for this cohort are already systematically reported and analyzed in Table 1; we therefore use accuracy in Table 2 to provide a clearer and more stable view of adaptation trends across methods.
>
> > **Q4:** Organization and table presentation
>
> We thank the reviewer for the helpful suggestions on table presentation and have updated Tables 1 and 2 accordingly in the revised version. Regarding organization, due to strict page limits, we prioritized core methodology and quantitative results in the main paper, and therefore placed some implementation details and qualitative results in the appendix. We nonetheless appreciate the reviewer’s constructive suggestion.
>
> > **Q5:** Contribution
>
> While the proposed aggregation module builds on existing graph attention mechanisms, it is designed to address aggregation instability observed in realistic WSI subtype classification settings. In addition to this targeted architectural design, a central contribution of this work is the systematic evaluation of graph-based WSI models under realistic and challenging conditions that are often underexplored in prior studies.
>
> Specifically, we consider rare-disease cohorts, strong class imbalance, inter-slide heterogeneity, and cross-site zero-shot and few-shot adaptation, and analyze model behavior using class-wise performance and forward/backward transfer metrics. These experiments provide empirical insights into how graph-based aggregation behaves in practice, with ResGAT serving as a simple and robust instantiation that consistently performs well under these conditions.

---

> ### Author Response · Authors · 2026-02-02
>
> We thank the reviewer for the positive assessment of our experimental protocol and the additional studies in the revision. For clarity of framing, we note that the hybrid kNN graph construction is presented as one component of ResGAT, while the contribution statement emphasizes the overall aggregation framework together with a systematic evaluation of graph-based WSI models under realistic and challenging settings (e.g., rare cohorts, imbalance, and cross-site scenarios). We hope this helps align the final framing with the contribution statement in our response.
>
> Regarding the comment that the empirical analysis of the hybrid kNN strategy is “mainly assessed in a single table,” we would like to note that, in addition to the comparison table between spatial-only, feature-only, and hybrid graphs, the revision includes a grid search over parameters, visualized as performance heatmaps (Figure 2 in the revised paper). These experiments show that ResGAT maintains stable performance across a broad range of configurations and motivate the “generalist” setting adopted in the main paper. Furthermore, we did add a computational efficiency analysis (Table 9 in the revised paper), reporting average training throughput (graphs/second, under identical hardware and batch settings) and showing that the multi-layer residual GAT design does not introduce significant overhead compared with WiKG and Patch-GCN. We fully agree that even more in-depth analyses of training stability and robustness are interesting directions, and we consider them valuable future work building on the current study.

---

### Official Review · Reviewer_1RPk · 2026-01-09

**Confidence:** 3
**Preliminary Rating:** 5
**Final Rating:** 5

**Summary:**

The paper proposes a graph-based method to solve the problem of classifying at the slide level on gigapixel images, too big to be processed as a single image. The method operates on patch graphs and models representations with stacked residual graph attention blocks. The method is a residual graph attention network. The evaluation is a binary subtype classification task across a class-imbalanced condition and two public TCGA datasets.

**Strengths:**

The paper is overall very well written, with a clear structure and an appropriate use of language, which makes it accessible and easy to follow. Despite the limited space available, the authors provide an informative and well-contextualized overview of the state of the art, allowing the reader to understand the motivation and positioning of the proposed approach. I particularly appreciate the inclusion of a dedicated evaluation to test cross-site generalization performance, as this is both highly relevant for real-world deployment and often overlooked in related work. The authors also include qualitative heatmap visualizations that help interpret the model’s behaviour and provide additional intuition beyond aggregate metrics. Finally, the paper compares the proposed method against a meaningful set of competing approaches, which strengthens the validity of the experimental conclusions and highlights the potential of the method within the current landscape.

**Weaknesses:**

The main weakness concerns the limited number of subjects available for training and evaluation. Although images are large and decomposed into many patches, the effective patient-level sample size remains small (e.g., 92 cases in the Appendiceal dataset and only a few hundred in TCGA). For heterogeneous histopathology tasks, this raises questions about robustness and generalizability, as patch-level augmentation cannot fully capture inter-patient variability. External validation or larger cohorts would help substantiate the empirical claims.

**Detailed Comments:**

Please provide more information on the pretrained encoder in the main text (at least the citation).

**Justification Of Final Rating:**

The paper includes more heatmaps and a better description of the method. Additionally, the requests from other reviewers appear to have been addressed. For next time, please highlight the modified text in a different colour.

**Justification Of The Preliminary Rating:**

Overall, the paper is well written, technically sound, and shows promising results. While the limited patient-level sample size raises some concerns about generalizability, the contribution remains valuable and justifies an acceptance.

**Questions To Address In The Rebuttal:**

Other than the point above, a few additional Heatmaps could be quite informative.

---

> ### Author Response · Authors · 2026-01-25
>
> We sincerely thank the reviewer for the positive and thoughtful evaluation of our work, and for highlighting the relevance of cross-site generalization, which closely aligns with the core motivation of this study.
>
> Regarding the pretrained encoder, we agree that this information should be made more explicit. In the revised paper, we have added the corresponding citation and a brief description of the pretrained feature extractor in the main text. We also appreciate the suggestion to include additional heatmap visualizations, and have provided more qualitative examples in the appendix.
>
> Finally, we agree that external validation or evaluation on larger cohorts would further strengthen the empirical conclusions. We view this as an important direction for future work to more broadly characterize model behavior and applicability across diverse pathology settings. In the present study, we focus on the proposed experimental setting and datasets.

---

### Author Rebuttal · Authors · 2026-01-25

**Rebuttal:**

We thank the reviewers for their thoughtful comments and suggestions, and thank the Area Chair for overseeing the review process.

In this revised submission, we preserve the overall narrative and core contributions of the paper, while addressing the reviewers’ scientific concerns through additional experiments and analyses. A subset of these additions has been partially incorporated into the main paper where they directly support the primary claims, while the majority of newly added experimental results, figures, and tables are currently included in the appendix. For several of these additions, we primarily provide the corresponding visualizations or tables and describe their motivation and implications in the rebuttal responses. We apologize that, owing to the limited rebuttal time window, the presentation and placement of these new results—including those appearing in the main paper—have not yet been fully streamlined, and will be comprehensively integrated in the final version.

**Supporting Material:**

/attachment/09131c8e7cd156114a2deaafeeec5f8c5bdc60d8.zip

---

### Meta-Review · Area_Chair_udM5 · 2026-02-04

**Recommendation:** Accept (Oral)
**Confidence:** 5

**Metareview:**

Following the rebuttal, the paper received one strong accept, one weak accept, and one weak reject. The authors provided a detailed and thoughtful rebuttal, incorporating additional experiments, analyses, and improved visualizations that addressed many of the previously raised concerns in the revised manuscript. One reviewer explicitly increased their score to weak accept after the rebuttal.

---

### Decision · Program_Chairs · 2026-02-13

Accept (Poster)